# Cognitive Flexibility in Hospitalized Patients with Severe or Extreme Anorexia Nervosa: A Case-Control Study

**DOI:** 10.3390/jpm13061000

**Published:** 2023-06-15

**Authors:** Simone Daugaard Hemmingsen, Nicolaj Daugaard, Magnus Sjögren, Mia Beck Lichtenstein, Claire Gudex, Frederikke Piil, René Klinkby Støving

**Affiliations:** 1Centre for Eating Disorders, Odense University Hospital, 5000 Odense, Denmark; shemmingsen@health.sdu.dk (S.D.H.); fpi@hejmdal.dk (F.P.); 2Research Unit for Medical Endocrinology, Odense University Hospital, 5000 Odense, Denmark; 3Research Unit, Child and Adolescent Psychiatry, Mental Health Services in the Region of Southern Denmark, 5000 Odense, Denmark; 4Department of Clinical Research, University of Southern Denmark, 5000 Odense, Denmark; mlichtenstein@health.sdu.dk (M.B.L.); cgudex@health.sdu.dk (C.G.); 5Open Patient Data Explorative Network (OPEN), 5000 Odense, Denmark; 6Department of Molecular Medicine, University of Southern Denmark, 5000 Odense, Denmark; nda@sdu.dk; 7Institute for Clinical Science, Department of Psychiatry, Umeå University, 901 87 Umeå, Sweden; magnus.sjogren@umu.se; 8Centre for Digital Psychiatry, Region of Southern Denmark, 5000 Odense, Denmark

**Keywords:** cognitive flexibility, anorexia nervosa, eating disorder, cognitive performance, neuropsychology

## Abstract

Objective: To investigate whether cognitive inflexibility could be identified using the Wisconsin Card Sorting Test (WCST) in patients with severe and extreme anorexia nervosa (AN) compared to healthy control participants (HCs). Method: We used the WCST to assess 34 patients with AN (mean age: 25.9 years, mean body mass index (BMI): 13.2 kg/m^2^) 3–7 days after admission to a specialized nutrition unit and 34 HCs. The Beck Depression Inventory II and the Eating Disorder Inventory 3 were distributed. Results: The patients displayed more perseveration than HCs controlled for age and years of education, with moderate effect sizes (perseverative responses (%): adjusted difference = −7.74, 95% CI: −14.29–(−1.20), *p*-value: 0.021; perseverative errors (%): adjusted difference = −6.01, 95% CI: −11.06–(−0.96), *p*-value: 0.020). There were no significant relationships between perseveration and depression, eating disorder symptoms, illness duration, or BMI. Discussion: Patients with severe and extreme AN demonstrated lower cognitive flexibility compared to HCs. Performance was not related to psychopathology or BMI. Patients with severe and extreme anorexia nervosa may not differ from less severe patients in cognitive flexibility performance. As this study exclusively focused on patients suffering from severe and extreme AN, potential correlations might be masked by a floor effect.

## 1. Introduction

Anorexia nervosa (AN) is a serious mental disorder characterized by a distorted body perception, fear of weight gain, and restricted food intake. AN has the highest mortality of all psychiatric disorders [1]. Patients with AN have an overall standardized mortality ratio of 5.9, but for severely malnourished patients with AN, this ratio is up to 15.9 [2]. Unfortunately, no clearly effective treatment exists for adults with AN [3], and patients with severe and enduring AN “present with one of the most challenging disorders in mental health care” [4]. 

The Diagnostic and Statistical Manual of Mental Disorders, Fifth Edition (DSM-5) presents a severity classification of AN that is based on body mass index (BMI). According to the DSM-5, patients with BMI between 15 and 15.99 kg/m^2^ are categorized with severe AN, whilst a BMI below 15 kg/m^2^ reflects extreme AN [5]. However, there is no consensus on severity definitions for AN in the literature. The International Statistical Classification of Diseases and Health-Related Problems, 11th revision (ICD-11) [6] presents two severity categories: AN with significantly low body weight (BMI 18.5–14.0 kg/m^2^) and AN with dangerously low body weight (BMI < 14.0 kg/m^2^). Other definitions of severity are based on illness duration instead of BMI. According to Broomfield et al., the most used definition of ‘severe and enduring AN’ in the literature was an illness duration of 7 or more years [7]. Patients with a very low BMI or a severe, fast weight loss often need somatic stabilization in a hospital. The current paper focuses on patients in need of somatic and medical stabilization with severe or extreme AN according to the DSM-5.

Malnutrition causes biological and endocrinological changes in individuals with AN, including amenorrhea [8]. The cognitive consequences of severe malnutrition are less clear [9], and the effect of severe food restriction on cognitive functions has been debated for years. The psychological and physiological consequences of food restriction were examined in 36 healthy men (conscientious objectors) in the Minnesota Starvation Experiment in 1944–45 [10]. The men were on severe food restriction for six months and lost 25% of their bodyweight. The researchers observed psychological changes that were similar to symptoms of AN as well as cognitive changes (including loss of concentration) but did not investigate neuropsychological performance. Newer research investigating cognitive consequences of restricted food intake in healthy individuals has only focused on short-term fasting (3–48 h) [11]. Preliminary evidence from a systematic review [11] suggested that short-term fasting is associated with executive function deficits. The neuropsychological consequences of long-term food restriction in healthy individuals remain unknown.

Similarly, the effect of malnutrition on cognitive functions in patients with AN are unknown. The investigation of this would require premorbid, prospective longitudinal studies of cognitive performance, e.g., birth cohorts. Studies of cognitive performance in patients with AN have therefore used cross-sectional designs that compared patients with AN to healthy participants or longitudinal designs that investigated cognitive improvement following treatment in patients with AN [9]. Studies have reported mixed results for the association between weight status and cognitive performance in patients with AN. Some found no association between weight and cognitive performance [12], whereas others found that lower weight was associated with lower cognitive performance [13]. 

Several cross-sectional studies investigating the executive function of cognitive flexibility found that patients with AN had lower performance than healthy individuals [14]. Cognitive rigidity (inflexibility) has also been suggested as a maintenance factor [15] that negatively influences engagement in treatment [16] and contributes to treatment resistance in patients with AN [17]. It has been discussed whether observed cognitive flexibility deficits in patients with AN compared to healthy individuals represent a trait or endophenotype in patients with AN [18] or whether the deficits are related to the state of the illness (a result of malnutrition) [19]. The former (the trait theory) is supported by research that found cognitive flexibility deficits in recovered individuals [20] and in relatives of individuals with AN, suggesting it could be an inherited trait [21]. The latter (the state theory) is supported by research that found age differences in cognitive flexibility performance [22], including that children and adolescents with AN have normal cognitive flexibility compared to healthy individuals [23,24,25] while adults with AN have cognitive flexibility deficits [22]. The state theory is further supported by the research reporting lower cognitive flexibility after fasting in healthy individuals (described above). If cognitive inflexibility is in fact a consequence of malnutrition in individuals with AN, then persisting lower performance following recovery could be interpreted as a scar effect.

The Wisconsin Card Sorting Test (WCST) is considered a gold standard measure of cognitive flexibility. Several studies investigating cognitive flexibility with WCST in adults with AN found that they performed worse than healthy control participants on perseverative errors, but others found no significant differences [14]. A recent systematic review of severe and enduring AN (illness duration of at least seven years) identified two studies and a case report investigating cognitive flexibility with the WCST in severe and enduring AN and also found mixed results [26]. Small sample sizes in several studies might explain the mixed results. Some of the differences may also be explained by nutritional conditions such as dehydration, vitamin and mineral deficiency, or low energy intake instead of weight status or BMI level. Not all studies of cognitive performance in hospitalized patients with AN reported the timing of assessments in relation to admission [9].

Although a considerable number of studies have investigated cognitive flexibility in patients with AN, few have used the WCST in samples of severely malnourished patients with AN [14,26]. Only one study [27] investigated patients with mean BMI under 14 prior to the current study. It remains unknown whether cognitive flexibility is related to degree of malnutrition in AN and whether severely malnourished patients with AN in need of medical stabilization display cognitive inflexibility.

## 2. Aims

The current study aimed to examine cognitive flexibility in severely malnourished patients with severe and extreme AN. The main objective was to investigate differences in perseveration performance on the WCST between patients with severe and extreme AN and healthy controls. Furthermore, we wanted to investigate relationships between perseveration performance and depression symptoms, eating disorder symptoms, illness duration, and BMI in the patients with AN.

## 3. Materials and Methods

This case-control study was nested in a cohort study that investigated cognitive performance, depression, anxiety, and cortisol levels in patients with severe and extreme AN before and after intensive nutritional and somatic stabilizing treatment during hospitalization [28,29]. Data collection was performed between March 2016 and January 2023. We used the secure web-based platform Research Electronic Data Capture (REDCap) [30] hosted at Odense Patient data Explorative Network (OPEN) for storing data and for collection of self-report questionnaires.

### 3.1. Inclusion of Patients with Anorexia Nervosa

The patients with severe and extreme AN were hospitalized in a specialized unit—the Nutrition Unit at the Center for Eating Disorders at Odense University Hospital, Denmark. Patients with life-threatening low weight are admitted to the inpatient Nutrition Unit for somatic, medical, and nutritional stabilization. For the current study, we included patients of any gender aged 16 years or older who were inpatients at the Nutrition Unit for at least three days and had been diagnosed according to the ICD-10 [31]. These patients were also diagnosed with AN according to the DSM-5 [5]. We initially invited 54 patients to participate in the study.

The subsequent exclusion criteria for study patients were active substance abuse, schizophrenia or other psychotic disorders, or short-term hospitalization (one to three days). Measurements were conducted 3–7 days after hospital admission (except for one patient who was assessed 14 days after admission due to functional disability). The final sample consisted of 34 patients who completed the assessments. 

### 3.2. Inclusion of Healthy Control Participants (HCs)

Medication-free HCs were included from the webpage forsoegsperson.dk (for recruitment of research participants), the local university, and a local high school. Possible participants were encouraged to send an e-mail with their age and educational level. Participants who matched the patients with AN on sex and were comparable in relation to age and educational level were invited to participate in the study. 

The exclusion criteria for the healthy participants were neurological illness, mental illness, previous eating disorder, and prescribed medication (oral contraception excluded). We included 37 HCs. Of these, 35 completed the assessments, and one was excluded. The final sample consisted of 34 matched HCs who were recruited between November 2021 and January 2023. 

### 3.3. Measurements

#### 3.3.1. Clinical Assessment

Clinical assessments included measurements of height (on a wall-mounted scale) and weight (on a calibrated platform) collected by specialized nurses at the Nutrition Unit. Demographic and clinical information (e.g., nadir BMI) for the patients with AN was collected from hospital records. 

#### 3.3.2. Self-Report Assessment of Depression and Eating Disorder Symptoms 

We distributed two self-report questionnaires in Danish to both patients and HCs: the Beck Depression Inventory (BDI-II) [32] and the Eating Disorder Inventory 3 (EDI-3) [33]. Both tools are widely used in research and clinical settings.

The BDI-II consists of 21 items and assesses the severity of depression symptoms. It was validated on a sample of patients with AN [34] and translated into Danish (Pearson Assessment, Danish version, 2005). Each item presents four statements scored on a scale from 0 to 3, and the participants must choose which statement best describes their thoughts, feelings, or behavior during the past two weeks. Higher scores represent more severe depression symptoms. Interpretation of severity: scores between 0–13 reflect no or minimal depression; scores between 14–19 reflect mild depression; scores between 20–28 reflect moderate depression; and scores between 29–63 reflect severe depression. 

The EDI-3 assesses symptoms associated with eating disorders. It was validated in Danish in patients with eating disorders [35]. The tool consists of 91 items rated on a 6-point Likert scale (‘always’ to ‘never’) and transformed into scores between 0–4. The items are organized into 12 subscales. The three subscales of Drive for Thinness, Bulimia, and Body Dissatisfaction were used in the current study. 

#### 3.3.3. Wisconsin Card Sorting Test

The WCST is a neuropsychological test that assesses abstract reasoning and cognitive flexibility [36] and is considered a gold standard measurement of executive function [37]. The test consists of 128 response cards (two decks) and four stimulus cards that depict figures of different forms (circle, cross, star, or triangle), different colors (blue, red, green, or yellow), and different numbers (one, two, three, or four) [36]. The participant is asked to match each of the cards in the deck of response cards to one of the four stimulus cards. The participant is told whether each response card was placed correctly or incorrectly. During the assessment, the sorting principle changes between form, color, and number, but the participant is not made aware that the sorting principle will change during the assessment. The assessment in the current study was administered by a psychologist (SH and FP). Raw scores for perseveration performance, general performance, and conceptual performance were calculated. The primary outcome was perseveration (percent perseverative responses and errors). 

### 3.4. Statistics

#### 3.4.1. Primary Outcomes

Percentages of the outcomes were calculated from the raw data, and afterwards, linear regression analyses were used to investigate differences in perseverative responses (%) and perseverative errors (%) between patients with AN and HCs, controlling for age and education in years (primary outcomes). Higher scores indicated more perseveration (and lower cognitive flexibility performance). We used a Šidák-corrected significance level, α = 0.025. Cohen’s d effect sizes were calculated. 

We included age and educational level in the statistical analyses as potential confounders because these have previously been identified as confounders in relation to cognitive flexibility performance measured with the WCST [36]. 

#### 3.4.2. Secondary Outcomes

Linear regression analyses were used to investigate whether patients with AN and HCs differed in other WCST raw scores (adjusted for age and education in years) and perseverative standard scores. The Šidák-corrected significance level was α = 0.006. Cohen’s d effect sizes were calculated. 

Correlations between percent perseverative responses and other variables (BDI-II-subscales, EDI-3 subscales, illness duration, and BMI) were analyzed with Spearman’s correlation (rho) since the variable perseverative responses (%) were non-normally distributed. The Šidák corrected significance level was α = 0.009.

Mean differences between groups (patients with AN vs. HCs) in Table 1 were analyzed with independent samples *t*-test.

#### 3.4.3. Sample Size Calculation

We expected a difference of seven points between patients with AN and HCs and a standard deviation of 10 on percent perseverative responses according to previous research [20]. With a significance level of 5% and the inclusion of 68 participants, we could achieve a power of 81%. We included 68 in total. 

### 3.5. Ethics

The study was approved by the Regional Research Ethics Committees of Southern Denmark (ID S-20150042). It was registered at clinicaltrials.gov accessed on 10 April 2023 (ID NCT02502617). Participants provided written informed consent.

## 4. Results

### 4.1. Descriptive Characteristics

Table 1 presents descriptive characteristics of the patients with AN and the HCs. A total of 66 females and two males aged 16 to 42 years were included in the study.

The mean BDI-II score for the patients with AN was in the category of severe depression.

### 4.2. Performance Differences between Patients and Healthy Participants

The subgroup variable (patients with AN versus HCs) was a statistically significant predictor of percent perseverative responses (Adj. difference = −7.74, 95% confidence interval (CI): −14.29–(−1.20), *p*-value: 0.021) and percent perseverative errors (Adj. difference = −6.01, 95% CI: −11.06–(−0.96), *p*-value: 0.020) when correcting for age and years of education. The effect sizes were moderate (0.57 and 0.58). For the patients with AN, the mean standard score (in relation to normative data from the test manual) of percent perseverative responses was 105.06 (standard deviation (SD): 31.14) and of percent perseverative errors was 102.76 (SD: 28.86)—this was interpreted as average performance according to the test manual (Heaton et al., 1993). For the HCs, the mean standard score of percent perseverative responses was 111.23 (SD: 21.56) and of percent perseverative errors was 108.94 (SD: 21.14)—this was interpreted as above-average performance.

Table 2 presents performance on the WCST measurements for patients with AN and the HCs as well as the adjusted differences between the groups. 

Within the group of patients with AN, there were no significant relationships between perseverative responses (%) and BDI-II scores (rho: −0.2406, *p*-value: 0.1704), EDI-3 subscales (drive for thinness: rho: −0.3751, *p*-value: 0.0288; bulimia: rho: −0.0529, *p*-value: 0.7665; body dissatisfaction: rho: −0.1057, *p*-value: 0.5521), illness duration (rho: 0.1316, *p*-value: 0.4582), or BMI (rho: 0.0934, *p*-value: 0.5994).

## 5. Discussion

The patients with severe and extreme anorexia nervosa demonstrated lower cognitive flexibility performance (higher percentages of perseverative responses and errors) than healthy controls on the WCST. However, cognitive flexibility did not seem to be related to psychopathology or weight status in the patients.

Perseveration and rigidity may be part of the AN disorder. Apart from being a possible maintenance factor of AN, cognitive inflexibility has been suggested to be related to lack of illness insight [38] and to implicate decision-making [39]. Cognitive flexibility may be a vital component of problem-solving skills in everyday life. The fact that the patients with severe and extreme AN in the current study displayed more perseveration than HCs may have implications, e.g., in everyday life and in psychological treatment settings. 

Our results confirm the results from previous studies comparing cognitive flexibility performance using the WCST between less severely malnourished adults with AN and HCs [20,40]. The systematic review by Miles et al. (2020) reported on three cross-sectional studies that investigated WCST performance in patients with acute AN compared to weight-restored and/or fully recovered patients. The studies found no statistically significant differences between groups, suggesting that weight status may not affect cognitive flexibility performance in AN [14]. In the current study, weight status seemed not to be related to perseveration for patients with AN with very low BMI (mean: 13.17, SD: 2.15). Although lower cognitive flexibility was not related to lower weight in the current study, the lower cognitive flexibility in patients with AN compared to HCs may still represent a consequence of malnutrition or other factors associated with the state of the illness (the state theory explained in the introduction). If this is the case, the cognitive inflexibility observed in recovered patients with AN in previous research may represent a scar effect. However, it may also be that lower cognitive flexibility is a trait in patients with AN that existed prior to the disorder and malnutrition (the trait theory). The discussion on trait or state-related factors is not specific to cognitive performance, however. Other circumstances such as the hypercortisolemia observed in patients with AN [41] might also represent either a persistent trait or be related to the state of the illness.

It has been debated whether statistically significant lower neuropsychological performance is the same as clinically significant lower performance [42]. Traditionally, neuropsychological performance of two standard deviations below the norm has been considered clinically significant [42,43], but one standard deviation below the norm has been suggested as clinically significant “cognitive underperformance” even when a score is within the normative range [44]. The (moderate) effect sizes in the current study did not exceed one standard deviation, a result in line with previous research [20,42]. This indicates a lower performance (underperformance) for the AN group but perhaps without cognitive flexibility impairment, as discussed in the systematic review by Stedal et al. (2021). It can be discussed whether the underperformance of cognitive flexibility found in the current study is clinically relevant. Compared to normative data for age and educational level from the test manual [36], the patients in the current study showed an overall average performance while the HCs demonstrated above-average performance. However, the normative data were derived from a sample from a different country (the United States) and were published 30 years ago in 1993. The lower performance (and tendency towards cognitive inflexibility) compared to the HCs in the current study could still have implications for the patients, e.g., by influencing decision-making or problem-solving, and they should be addressed in treatment. 

The results of the current study highlight the importance of comparing results for patients with AN with those for a healthy control group. Perseveration was not normally distributed in the group of patients with AN, and one subgroup of patients with severe and extreme AN can display impaired cognitive flexibility while another subgroup may show normal or good cognitive flexibility performance. A study of enduring AN also found increased accuracy performance in patients compared to HCs [45]. These findings support the need for individual evaluation and assessment of cognitive flexibility in patients with severe and extreme AN as appropriate. Information about whether a patient performs well or has cognitive impairment may be relevant in a therapeutic setting and could help health care professionals to tailor treatments to individual patients. 

It is possible that cognitive performance is related to depressive symptoms in patients with AN as depression is a well-known comorbid condition [46,47]. People with depression seem to have lower cognitive performance compared to healthy individuals [48] but in the current study, severity of depressive symptoms was not related to level of cognitive performance. Our sample consisted primarily of patients with extreme AN and severe depressive symptoms, however, and the inclusion of less severe patients in future studies might help to reveal correlations between depressive symptoms and cognitive flexibility. A recent study suggested that rumination, perfectionism, and cognitive inflexibility are interrelated in patients with AN [49]. 

An investigation of self-perceived cognitive flexibility in everyday life might have produced different results to the current study of neuropsychological functioning in hospitalized patients. Miles et al. (2023) found that although patients with AN had similar cognitive flexibility on the WCST to healthy individuals, their self-perceived cognitive flexibility was significantly lower. This might suggest that even slightly poorer performance on a neuropsychological test such as the WCST can be associated with everyday difficulties for individuals with severe AN. Patient characteristics may be important, however, as the patients in the study by Miles et al. had longer mean illness duration (mean: 10.4 years, SD: 8.47) and higher weight status (mean BMI: 16.88, SD: 2.93) than our sample of patients with severe and extreme AN.

Extremely malnourished patients with AN are not well described in the literature as a minority of patients reach this degree of severity of AN. Few specialized somatic units treat this patient group and many patients with extreme AN with life-threatening low weight are treated in endocrinological departments that have few patients with AN at any one time. The current sample included a case of a 35-year-old woman with a BMI of 7.7 who performed remarkably well on many cognitive functions but displayed cognitive impairment measured on the WCST [29]. Although our study results confirmed our hypothesis that the patients with severe and extreme AN would display lower cognitive flexibility than healthy individuals, we could not confirm a cognitive flexibility of two standard deviations below the norm. It appears, therefore, that patients with severe and extreme AN do not differ markedly from patients with less severe AN in relation to cognitive flexibility performance. Newly diagnosed patients may differ from patients with enduring AN on cognitive flexibility performance, regardless of weight status, which calls for further investigation in future studies. 

### 5.1. Strengths and Limitations

We conducted assessments after at least three days of hospitalization, after fluid and electrolyte correction and acclimatization. This is a strength of the study as factors such as dehydration, vitamin and mineral deficiency, hypoglycemia, and low energy intake may influence cognitive performance.

It is a limitation of the study that the sample of HCs was not matched one-to-one to the patients with AN on age and educational level. Several patients dropped out of primary school or high school due to their eating disorder, and it was difficult to find HCs in the community who matched the AN group on both parameters as well as sex. We therefore corrected for age and educational level in the statistical analyses, and matched one-to-one on sex. In addition, we would have preferred the sample to be larger. However, due to the severity of the patients, it took several years to include the current sample of 34 patients.

### 5.2. Conclusions

The patients with severe and extreme AN demonstrated lower cognitive flexibility performance (more perseveration) than healthy individuals. However, psychopathology and weight status were not associated with perseveration for the patients. Since this study exclusively focused on patients with severe and extreme AN, potential correlations might be masked by a floor effect. However, the results suggest that patients with severe and extreme AN may not differ markedly from less severely malnourished patients in cognitive flexibility performance. Cognitive inflexibility could be addressed in psychotherapeutic treatment when patients are malnourished during hospitalization.

Future studies that include samples of adult patients with mild, moderate, severe, and extreme AN could investigate whether higher BMI is related to better cognitive flexibility performance in adults with AN. Since perseveration was not normally distributed in the group of patients with severe and extreme AN, it could also be investigated whether one subgroup of patients with severe and extreme AN has cognitive flexibility impairment while another subgroup has good cognitive flexibility performance. This would require a larger sample of patients with severe and extreme AN and therefore could be a multicenter study. In addition, it would be useful to investigate self-reported cognitive flexibility in patients with severe and extreme AN, since perceived cognitive flexibility may differ from the results of neuropsychological investigations.

## Figures and Tables

**Table 1 jpm-13-01000-t001:** Descriptive characteristics of the patients with severe and extreme anorexia nervosa (AN) and the healthy control participants (HC).

	AN		HC		Groups Differences
	n (%)	Mean (SD)	n (%)	Mean (SD)	*p*-Values
**Total**	34 (100)		34 (100)		
**Sex**					
Female	33 (97)		33 (97)		
**Age, years**		25.91 (7.53)		23.15 (2.46)	0.046
**Education, years**		11.91 (2.23)		13.71 (1.71)	<0.001
**BMI, kg/m^2^**		13.17 (2.15)		22.84 (2.47)	<0.001
**Questionnaires**					
**BDI-II**		34.18 (10.85)		5.82 (6.25)	<0.001
**EDI-3 subscales**					
Drive for thinness		18.88 (8.06)		2.76 (3.84)	<0.001
Bulimia		7.06 (8.06)		2.41 (3.51)	0.003
Body dissatisfaction		27.35 (11.08)		7.06 (7.10)	<0.001
**Patient characteristics**					
**Nadir BMI, kg/m^2^**		12.11 (2.15)			
**Illness duration, years**		7.94 (6.24)			
**AN severity, DSM-5 categories**					
Severe (BMI ≧ 15)	6 (18)				
Extreme (BMI < 15)	28 (82)				
**Diagnostic subtype**					
Restrictive	20 (59)				
Binge/purging	14 (41)				
**Co-morbidity**					
None	26 (76)				
Obsessive compulsive disorder, personality disorder, or autism spectrum disorder	8 (24)				

SD = standard deviation; BMI = body mass index; BMI-II = Beck Depression Inventory II; EDI-3 = Eating Disorder Inventory 3; Nadir BMI = lowest BMI registered in hospital records; DSM-5 = Diagnostic and Statistical Manual of Mental Disorders 5th edition.

**Table 2 jpm-13-01000-t002:** Wisconsin Card Sorting Test performance raw scores and differences between the 34 patients with severe and extreme anorexia nervosa (AN) and the 34 healthy control participants (HC).

	AN Mean (SD)	HC Mean (SD)	Adjusted Difference (95% CI)	*p*-Values	Cohen’s d
**Perseveration performance**					
Perseverative responses (%) ^a^	16.99 (15.33)	10.28 (6.18)	−7.74 (−14.29–(−1.20))	0.021 *	0.5741
Perseverative errors (%) ^a^	14.91 (11.71)	9.66 (5.12)	−6.01 (−11.06–(−0.96))	0.020 *	0.5809
**General performance**					
Number of trials administered ^b^	98.38 (24.23)	84.82 (20.10)	−17.22 (−29.29–(−5.14))	0.006 *	0.6091
Total correct responses ^b^	69.03 (9.38)	66.85 (6.39)	−3.71 (−8.16–0.73)	0.100	0.2716
Errors (%) ^b^	30.91 (29.04)	18.61 (11.90)	−13.39 (−25.90–(−0.88))	0.036	0.5543
Non-perseverative errors (%) ^b^	11.07 (7.48)	8.80 (7.57)	−2.51 (−6.62–1.60)	0.226	0.3017
Categories completed ^b^	4.85 (1.86)	5.50 (1.44)	1.13 (0.25–2.01)	0.013	−0.3908
**Conceptual performance**					
Trials to complete first category ^b^	14.50 (6.99)	17.71 (22.68)	1.60 (−7.84–11.04)	0.736	−0.1913
Conceptual level responses (%) ^b^	64.63 (23.61)	77.10 (17.78)	14.67 (3.09–26.25)	0.014	−0.5967
Failure to maintain set ^b^	0.97 (1.95)	0.29 (0.84)	−1.04 (−1.86–(−0.22))	0.014	0.4529

SD = standard deviation, CI = confidence interval. ^a^ Primary outcomes (Šidák corrected significance level: α = 0.025). ^b^ Secondary outcomes (Šidák corrected significance level: α = 0.009). * Indicates significance at the corrected level. All the linear regression analyses were adjusted for age and years of education.

## Data Availability

The data that support the findings of this study are available from the corresponding author upon reasonable request.

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
