# Peer review of "Cognitive Flexibility in Hospitalized Patients with Severe or Extreme Anorexia Nervosa: A Case-Control Study"

_jpm, 2023, doi:10.3390/jpm13061000_

Round 1

Reviewer 1 Report

The manuscript submitted by Hemmingsen et al. presented the investigation result of an attractive yet long-disputed question of whether the apparent reduction of cognitive abilities in severe and extreme AN patient was trait-based or state-based. Also, we could observe correlations between severe/extreme AN status, cognitive abilities, and clinical traits such as BMI in either case. Whilst the sample size was relatively small, and the distribution of samples was essentially binary, they successfully reached some initial conclusion that, whilst a correlation between severe/extreme AN status and cognitive abilities could be established, the data did not support that a correlation between cognitive abilities and clinical traits in severe/extreme AN patients. Nevertheless, the answer to the trait/state question remained elusive, primarily due to the binary grouping method used in the study (which is practically restricted by the small patient count and limited availability of patients of different tiers), which calls for further investigation in the studies that may follow up.

The manuscript is generally well-written, with a few minor grammar glitches between lines 300-326 (probably written by another author?). It would be great if those paragraphs get further polished before publication.

Author Response

We would like to thank the Editor and the Reviewers for their time end effort to review our manuscript. We are very grateful for the opportunity to revise the manuscript, and we believe that the revisions made based on the reviewers’ valuable comments and suggestions have improved the quality of the manuscript. We have added answers to each comment below. 

Reviewer 1

Comment: The manuscript submitted by Hemmingsen et al. presented the investigation result of an attractive yet long-disputed question of whether the apparent reduction of cognitive abilities in severe and extreme AN patient was trait-based or state-based. Also, we could observe correlations between severe/extreme AN status, cognitive abilities, and clinical traits such as BMI in either case. Whilst the sample size was relatively small, and the distribution of samples was essentially binary, they successfully reached some initial conclusion that, whilst a correlation between severe/extreme AN status and cognitive abilities could be established, the data did not support that a correlation between cognitive abilities and clinical traits in severe/extreme AN patients. Nevertheless, the answer to the trait/state question remained elusive, primarily due to the binary grouping method used in the study (which is practically restricted by the small patient count and limited availability of patients of different tiers), which calls for further investigation in the studies that may follow up.

Answer: We want to thank the reviewer for this comment. We very much agree that this calls for further investigation in follow-up studies.

Comment: The manuscript is generally well-written, with a few minor grammar glitches between lines 300-326 (probably written by another author?). It would be great if those paragraphs get further polished before publication.

Answer: We agree. The paragraphs have been proofread by a native English speaker and changes are tracked in the manuscript at pp 11, lines 307-330.

Reviewer 2 Report

Anorexia is an eating disorder that causes people to weigh less than what is considered healthy for their age and height, usually due to excessive weight loss. People with this disorder may have an intense fear of gaining weight, even when they are underweight. It is important to carry out this type of study to identify treatments at an adequate time.

The following recommendations are sent to the authors for their consideration:

1. The authors mention that it is a cross-sectional study of cases and controls nested in a cohort. It is suggested to properly identify the type of study... either it is cross-sectional or it is a case-control study.

2. Were patients of any age and gender included?

3. It is suggested to place more information on how the anthropometric measurements were made to the patients. Who made the measurements?

4. Were patients with anorexia nervos included in the study regardless of the time of diagnosis? If so, could this modify the results?

5. The authors mention in the statistical analysis section that a regression analysis was used, however, it is not identified what type of regression it was.

6. The authors mention that the study was registered in clinical trials, however, the text only mentions that the data analyzed is nested in a cohort. So the study is nested in a clinical trial? The intervention carried out with the patients was not identified.

7. In Table 1 it is suggested to place the percentages that correspond to the categorical variables (sex, BMI categories, diagnosis... etc). Likewise, it is suggested to add a column with the p values of the differences between the study groups.

8. Table 2 shows that they obtained B values and the confidence interval and again the type of regression that was carried out is not mentioned, it seems that it is a linear regression, which is not adequate for this study since it is a cases and controls. It is suggested to carry out a logistic regression analysis.

9. On the other hand, it is suggested to generate a bivariate analysis with their respective p values to later generate a regression model.

Author Response

We would like to thank the Editor and the Reviewers for their time end effort to review our manuscript. We are very grateful for the opportunity to revise the manuscript, and we believe that the revisions made based on the reviewers’ valuable comments and suggestions have improved the quality of the manuscript. We have added answers to each comment below.

Reviewer 2

Anorexia is an eating disorder that causes people to weigh less than what is considered healthy for their age and height, usually due to excessive weight loss. People with this disorder may have an intense fear of gaining weight, even when they are underweight. It is important to carry out this type of study to identify treatments at an adequate time.

The following recommendations are sent to the authors for their consideration:

Comment 1: The authors mention that it is a cross-sectional study of cases and controls nested in a cohort. It is suggested to properly identify the type of study... either it is cross-sectional or it is a case-control study.

Answer 1: We want to thank the reviewer for pointing this out. We have revised the design to “case-control study” in the manuscript, pp 4, line 129.

Comment 2: Were patients of any age and gender included?

Answer 2: We invited patients of any gender, who were aged 16 years or older. We have added information on this in the manuscript:
For the current study, we included patients of any gender aged 16 years or older …”, pp. 5, line 139.
“A total of 66 females and two males aged 16 to 42 years were included in the study.”, pp. 7, lines 229-230.

Comment 3: It is suggested to place more information on how the anthropometric measurements were made to the patients. Who made the measurements?

Answer 3: We agree. The specialized nurses at the Nutrition Unit collected anthropometric measurements. We have added information on this in the manuscript at pp 5, lines 162-163.

Comment 4: Were patients with anorexia nervosa included in the study regardless of the time of diagnosis? If so, could this modify the results?

Answer 4: Yes, the patients were included few days after admission to a specialized somatic unit for eating disorders regardless of first onset of anorexia nervosa, and the sample was generally characterized by patients with enduring (or chronic) anorexia nervosa. The reviewer is making a very good point. We agree that onset of anorexia nervosa and illness duration may affect results regardless of weight status. We have added the following sentence in the manuscripts at the end of the discussion section:

Newly diagnosed patients may differ from patients with enduring AN on cognitive flexibility performance, regardless of weight status, which calls for further investigation in future studies” pp. 11-12, lines 330-683.

Comment 5: The authors mention in the statistical analysis section that a regression analysis was used, however, it is not identified what type of regression it was.

Answer 5: We thank the reviewer for pointing this out. We have revised the statistical analysis section to include that we used linear regression analyses, pp. 6, lines 195 and 204. Please, see Answer 8.

Comment 6: The authors mention that the study was registered in clinical trials, however, the text only mentions that the data analyzed is nested in a cohort. So the study is nested in a clinical trial? The intervention carried out with the patients was not identified.

Answer 6: We thank the reviewer for this comment. We have now specified that the current study is a case-control study nested in a cohort study investigating cognitive performance, cortisol concentrations, anxiety symptoms, and depression symptoms in patients with severe and extreme anorexia nervosa before and after nutritional and somatic stabilizing treatment during hospitalization. The intervention was intensive nutritional and somatic stabilization targeted life-threatening anorexia nervosa, and the cohort study was registered at clinicaltrials.gov, but it was not a randomized controlled trial. We have added the following in the manuscript:

“This case-control study was nested in a cohort study that investigated cognitive performance, depression, anxiety, and cortisol levels in patients with severe and extreme AN before and after intensive nutritional and somatic stabilizing treatment during hospitalization”,pp. 4, lines-129-130.

Comment 7: In Table 1 it is suggested to place the percentages that correspond to the categorical variables (sex, BMI categories, diagnosis... etc). Likewise, it is suggested to add a column with the p values of the differences between the study groups.

Answer 7: We have added the percentages for all categorical variables and the p-values of group differences in Table 1, pp. 8, line 235, as suggested. In addition, we have added the following to the statistical section in the manuscript:

“Mean differences between groups (patients with AN vs HCs) in Table 1 were analyzed with independent samples t-test.”, pp. 6, line 212.

Comment 8: Table 2 shows that they obtained B values and the confidence interval and again the type of regression that was carried out is not mentioned, it seems that it is a linear regression, which is not adequate for this study since it is a cases and controls. It is suggested to carry out a logistic regression analysis.

Answer 8: In addition to the revision of the statistical analysis section (see below), we have revised the note for Table 2, pp. 9, line 253, accordingly. The main aim of the study was to investigate differences in percentages of perseverative answers and errors between patients and controls. We calculated percentages of perseverative answers and errors from the raw scores and analyzed these continuous outcomes with linear regression analyses.

We have added type of regression analyses for primary outcomes in the statistical analysis section:

“Percentages of the outcomes were calculated from the raw data, and afterwards, linear regression analyses were used to investigate whether group affiliation (patients with AN or HCs) predicted perseverative responses (%) and perseverative errors (%), controlling for age and education in years (primary outcomes).”, pp 6, lines 195-197.

We have also added the statistical analyses used for the continuous secondary outcomes:

“Linear regression analyses were used to investigate whether subgroup (patients with AN or HCs) was a predictor of other WCST raw scores (adjusted for age and education in years) and perseverative standard scores.”, pp 6, lines 204-205.

Comment 9: On the other hand, it is suggested to generate a bivariate analysis with their respective p values to later generate a regression model.

Answer 9: We consulted a medical statistician and used linear regression analyses for the continuous outcomes. Please see Answer 8 above.

Round 2

Reviewer 2 Report

Table 2 is still not understood, despite the fact that a linear regression was carried out and that it is a case-control study, the table does not identify which group is the one that is being compared, for example, the value of B of the variable responses perserverantes is for the control group or with anorexia?

Author Response

We want to thank the Editor for the opportunity to revise the manuscript and the Reviewers for their time and valuable feedback.

We have addressed the comment from Reviewer 2 below.

Comment: Table 2 is still not understood, despite the fact that a linear regression was carried out and that it is a case-control study, the table does not identify which group is the one that is being compared, for example, the value of B of the variable responses perserverantes is for the control group or with anorexia? 

Answer: We want to thank the Reviewer for pointing this out. We used linear regression instead of t-test to examine differences between groups, because we wanted to adjust for age and years of education as potential confounders. The regression coefficient is the difference between the outcome for cases and the outcome for controls, adjusted for age and education (years). To make Table 2 easier to read, we have changed the column title to ‘Adjusted difference’ and revised the footnote for the table, page 9, line 258. Furthermore, we have added ‘adjusted differences’ in the text above Table 2, at line 255. In the statistical section at page 6, we have changed the description accordingly, lines 197 and 205. In addition, we have changed ‘B’ to ‘adjusted difference’ in the Abstract, page 1, lines 35-36, and at page 8, lines 245-246. We believe that this change throughout has made the manuscript easier understand for the reader and we sincerely thank the Reviewer for the comment.
